# Towards a New Urban Health Science

**Franz W Gatzweiler** [1,2,*] , **Saroj Jayasinghe** [3,4] , **José G Siri** [5] and **Jason Corburn** [6]

1   Institute of Urban Environment, Chinese Academy of Sciences, Xiamen 361021, China
2   United Nations University Institute in Macau, Macau SAR, China
3   Faculty of Medicine, University of Colombo, Kynsey Road, Colombo 00800, Sri Lanka
4   Faculty of Medicine, Sabaragamuwa University of Sri Lanka, Belihuloya 70140, Sri Lanka
5   Independent Researcher, Philadelphia, PA 19146, USA
6   School of Public Health & Department of City & Regional Planning, Center for Global Healthy Cities, UC Berkeley, Berkeley, CA 94720, USA
*   Correspondence: franz@unu.edu; Tel.: +853-6801-7217

**Abstract:** The intensity and range of health challenges that people in cities are facing has increased in recent years. This is due in part to a failure to adequately adapt and respond to emergent and expanding global systemic risks, but also to a still-limited understanding of the profound impacts of complexity on urban health. While complexity science is increasingly embraced by the health and urban sciences, it has yet to be functionally incorporated into urban health research, policy, and practice. Accelerating urbanization in a context of escalating environmental constraints will require deeper engagement with complexity, yet also, paradoxically, much swifter, more effective, and more risk-averse decision-making. Meeting these demands will require adopting a science, policy and practice style which is integrative, inclusive, collaborative, systemic, fast, and frugal. We propose transformational shifts in scientific methodology, epistemological and ontological stances, types of rationality, and governance to shift researchers, policymakers, practitioners, and citizens towards a new, complexity-informed science of urban health.

**Keywords:** complex systems; urban health; inclusion; knowledge; rationality; governance

## 1. Introduction

While urban health scientists and decision makers have long acknowledged that cities are complex systems, inquiry and practice have yet to fully contend with the implications. We suggest here that approaching cities as complex systems will require significant, explicit changes to current research and practices for urban health. Failing to take such an approach contributes to data, decisions, and outcomes that perpetuate inequities, reduce freedoms, fail to adequately address environmental and ecosystem degradation, and contribute to suboptimal or declining health for the billions living in urban areas—particularly in times of global health crises, like those arising from climate change and the recent COVID-19 pandemic. As such, urban health should pursue a deeper, more pragmatic engagement with urban complexity science. This would help address the most urgent health and equity issues in cities, reduce the unintended consequences of urbanization—and of sustainable development itself—and strengthen competencies for governing complex urban systems.

Over the last several decades, various commissions and commentaries have called for more research on how city decision making and urban design influence population health and health equity, and for corresponding action [1–3]. These important reviews themselves build upon an older literature demanding that urban design, planning, and policy recognize the unique characteristics that contribute to well-being or ill health in urban settings [4–7]. This half-century of incremental work holds critical insights for urban health, yet there is still considerable scope to benefit from a new perspective on the interface between urban health science and policymaking that:

- acknowledges and prioritizes persistent inequities and recognizes their substantial origins in urban complexity;
- proceeds from a basic recognition that urban health is an emergent outcome of complex processes, and accordingly interrogates linear assumptions about health causation;
- adopts a multiscale perspective, considering how complex regional and global connectedness, networks, and flows impact urban health;
- focuses on ways to foster civic participation in urban goal setting, implementation, governance, monitoring, and communication;
- recognizes and works toward local resilience as an enabling framework for anticipating, responding to, and learning from health crises;
- places a greater emphasis on history and learning, especially from failed urban health experiments and under-appreciated successes;
- makes use of new digital tools and techniques.

Urban health scholars and practitioners have already, in many contexts, recognized that delivering more healthy, equitable, and inclusive cities depends on engaging more substantially with urban complexity, rather than attempting to simplify the irreducible. In fact, it is increasingly accepted that many negative health impacts in cities emerge due to the difficulty of perceiving, understanding, and governing urban complexity [8].

Important advances have been made in polycentric, participatory, reflexive, adaptive, and transdisciplinary research and policy styles, which can be viewed as attempts to engage with complexity and match complex decision-making situations with requisite governance institutions (e.g., structures, mechanisms, rules) [9]. Yet, recent times have also seen a reverse trend, towards more top-down/impositional and less bottom-up/consensual policy styles [10–12]; this trend has likely been accelerated by global health emergencies, risks, and disasters, and especially by the recent global COVID-19 pandemic [13].

This is not to say that more consensual policy styles are always 'better'. Rather, to effectively navigate an increasingly complex social, ecological, and technological urban environment, it is essential to have policy styles, participatory decision-making mechanisms, and governance capacities that match the constraints and opportunities imposed by complexity. The first law of cybernetics states that systems must have a number of control or response mechanisms equal to or greater than the number of potential disturbances the system faces [14]. Policymaking under complexity rarely obeys this law, usually due to the competing interest of creating more economically efficient outcomes.

Vatn [15] argues for the importance of matching policy regimes to complexity, noting that the latter conditions the very framework for decision choices. In contrast to simple modular systems, complex systems demand consideration of not only individual but social rationality, and not only instrumental but communicative rationality. In other words, they require that we consider not only the sum of progress toward individual goals but progress toward collective goals, and not only technical means of achieving outcomes but mechanisms that foster iterative, consensual understanding. Too often, institutional inertia causes science and policy to be based merely on individual rationality and instrumental types of human interactions, leading to cost-efficient but suboptimal choices and unintended negative consequences, particularly for complex issues such as urban sustainability and health. Mueller [16] traces "ubiquitous" policy failure to the false assumption that complex systems can be precisely determined, closely predicted and exactly controlled—a paradoxically simplistic acknowledgement of complexity that leads to overconfidence and unrealistic expectations.

Reservations against engaging more deeply with complexity are sometimes predicated on the perception that such engagement precludes timely and rigorous application of the scientific method. This is an especially poignant critique in the context of escalating challenges, where solutions are needed much faster than traditional science can typically provide them. Yet narrow decision-making based on linear assumptions often leads to calamitous outcomes in complex systems; in the words of Read [17] (often misattributed to Keynes), "It is better to be vaguely right than exactly wrong." Moreover, decision-

making under complexity need not be fatally slow: there is a rich history of rapid, heuristic decision-making successfully applied to complex situations in health [18,19].

Decision-making guided by complexity science may also be perceived as lacking clarity (e.g., of goals, methods, or messages) and can accordingly give rise to mistrust. This is one reason why solutions to complex urban health issues usually depend on collective action; involving all relevant stakeholders across all pertinent scales increases the legitimacy of decisions, even where complexity imposes irreducible ambiguities.

Decision-makers themselves can impose additional barriers to engaging with complexity. For example, they may choose to persevere with an existing policy style or decision-making approach to maintain credibility and avoid the perception of unreliability—a behavior explainable through the lens of so-called 'sunk costs'. Complexity, often confused with 'complicatedness', can engender resistance based on the perceived costs of change—especially when changes might threaten policymakers' accustomed roles and positions. Vested interests will often prevent decision makers from making necessary changes to systems from which they benefit. These factors apply even where existing outcomes are globally suboptimal, and a complexity-informed approach would benefit the common good.

The challenge of systematizing complexity thinking in research, policy and practice is substantial. It calls for institutionalizing the participatory processes of knowledge-making within institutions that have consistently worked to isolate expert knowledge from the vagaries of dissent, populism, and politics. In grappling with the partiality of scientific knowledge and the inevitable uncertainty associated with complex reality, Jasanoff [20] made the case for 'technologies of humility'; methods—or better yet, institutionalized habits of thought—that seek to come to grips with the ragged fringes of human understanding: the unknown, the uncertain, the ambiguous, and the uncontrollable. Acknowledging the limits of prediction and control, technologies of humility confront head-on the normative implications of our lack of perfect foresight. They call for different capabilities and different forms of engagement between experts, decision-makers, and the public than have previously been considered needful in governance. They require not only formal mechanisms of participation but also an intellectual environment in which citizens are encouraged to bring their knowledge and skills to bear on the resolution of common problems.

In sum, it has become clear that informational and epistemological frameworks for policymaking must be adjusted in the face of complexity. In this commentary, we propose modifications to policymaking styles for urban health, elaborating on how complexity science and policy can be applied to urban health challenges to meet the demands of the 21st century.

## 2. Toward a New Urban Health Science

There is a long history of understanding and engaging with cities as complex systems. In the mid-19th century, Cerda [21] laid the groundwork for a science of cities on the basis of observations of geometry, form, and notions of mechanism at work. Geddes [22], half a century later, perceived cities as evolving from flows and networks. More recently, alongside significant advances in the field of complexity science, cities came to be seen as products of bottom-up, evolutionary, self-organizing processes, rather than top-down design [23,24].

Over the past decade, a new urban science has emerged which recognizes that cities are centers of complexity, encompassing multiple types of embedded, overlapping, and interacting systems [25–27]. Because this complexity is partly organized and partly emergent, cities are partly plannable and partly unpredictable and unplannable, depending on the physical and time scales of observation.

Urban health, defined here as the health of people living in cities and the complex state of the environments on which it depends, is no less subject to complexity than are cities themselves. Accordingly, to continue to safeguard and improve human health and the environment, especially in the context of escalating risks, the field of urban health

will also have to come to grips with urban complexity. A new science for healthy cities would draw on lessons from the broader inquiry into urban complexity [28]. It would complement an understanding of urban health as the outcome of multiple social and environmental determinants, deepening the recognition of urban health as a product of the mutual interactions between humans and their environments—or, more broadly, as an emergent property of interacting socio-ecological-technological systems (SETS) [29,30].

The shift in focus from a multiple determinants perspective to one focusing on systemic interactions implies a corresponding change in the structure of the scientific enterprise; whereas the former can be studied by teams of researchers from multiple disciplines working in parallel, the latter requires consilience (converging evidence from many unrelated lines of inquiry) and the robust uptake of inter- and transdisciplinary inquiry [9,31,32]. This, in turn, requires a greater focus on participatory processes in order to ensure the multiple lines of evidence needed to generate new insights and inform policy and practice (among other benefits). In the words of Jane Jacobs [33], "Cities have the capability of providing something for everybody, only because, and only when, they are created by everybody."

By its nature, a complexity-informed science of urban health would encourage a focus on the upstream root causes of urban health and environmental challenges, allowing for a more efficient, proactive approach rooted in prevention, rather than the more typical reactive mode that endeavors to alleviate problems. Through its embrace of inclusive participation and a focus on systemic unintended consequences, it would also offer insights and incentives for addressing pervasive social, institutional, and health inequities in cities [34].

Critically, such an approach would also be better suited to understanding and finding solutions in the context of the characteristic hyperconnectivity of modern urbanization. As articulated by Batty [26], "In a world now dominated by communications . . . it is high time we changed our focus from locations to interactions, from thinking of cities simply as idealized morphologies to thinking of them as patterns of communication, interaction, trade, and exchange; in short, to thinking of them as networks." Urban spaces feature exponentially accelerating connectivity. For example, modern transport systems enable rapid and thus more geographically extensive mobility, and thus larger numbers of potential person-to-person connections. The dense, intensely interconnected, rapidly changing milieux enabled by transport and other urban systems require governance responses that are equally fast, flexible, diverse, and multi-source, as has been observed for the special case of COVID-19 [35,36].

Urban connectivity is not limited to person-to-person interactions. Through systemic connections, urban complexity translates human activity into broad impacts on people and environments, solving some problems but also creating new 'wicked' challenges, themselves inextricably connected [37]. Thus, global urban systems have driven growth and development, raised life expectancy, and reduced poverty. However, urban growth is founded on—and has systematically degraded—the global ecosystems to which global networks of cities are connected and which provide the resources to build and feed cities and the sinks which absorb their wastes [38,39].

Thus, the better understanding of the city as a complex system eventually leads us to a firmer appreciation of the deep connections between cities and our planet. In the face of increasingly systemic global risks [13] and the increasing likelihood that we will collectively fail to achieve the United Nations Sustainable Development Goals (SDGs) [40] by 2030 [41], this understanding may also lead us to crucial new insights into how to define and pursue sustainability, drawing on lessons from complexity science and on our capacity for collective action and intelligence. It should also foster engagement with existing conceptual critiques of sustainable development [42–44]—in particular the ecological incoherence of the most widely visible measures of 'progress' [45]—fostering better outcomes by expanding the discourse of ideas.

### 3. Shifts to Enable a Complexity-Informed Approach to Urban Health

Based on long-standing observations by scientists, urban residents and other stakeholders, and decision makers, we propose *four critical shifts* to underpin a complexity-informed approach to urban health. These shifts have to do with (a) how we perceive the nature of the challenges we face (ontology) and our ability to have knowledge about them (epistemology); (b) the tools we use to derive that knowledge (methodology); (c) the way that we respond to and make decisions based on that knowledge (rationality); and (d) the way we organize our institutions to promote such an approach (governance).

1. **Nature and how we know it: shifting ontological and epistemological stances**

For close to half a millennium, the scientific method has been the primary basis for our claim to "know" reality, and thus the basis for evidence-informed decision-making. Indeed, Wilson observed that "with aid of the scientific method, we have gained an encompassing view of the physical world far beyond the dreams of earlier generations" [46]. Yet, this expanded view, and indeed the scientific method itself, has in some contexts been challenged as failing to represent a full or useful picture of real-world problems.

As classically applied, the scientific method tends to assume that natural processes are reducible to observable, testable cause-and-effect relationships among independent variables. It tends toward quantitative rather than qualitative analysis, discourages ambiguity, and perceives the analyst as an objective, unboundedly rational individual standing 'outside' the observed system. In philosophy this belief has been referred to as realism.

Reality, however, involves a host of complications: non-linear feedback relationships, multiway causality, emergent behavior, and the methods by which we observe or try to discover realty. The nature of human processes means that scientific production itself is subject to complex biases. It is therefore rarely helpful to ask which model is closest to one universal reality. What is more useful is to acknowledge that all models fail to completely account for the complexities of reality and that some models are more useful than others: model-dependent realism [47].

Thus, classical science often fails to produce actionable knowledge or meet societal needs. Models fail to reflect people's lived realities. Research cycles are consistently outpaced by events, institutional mechanisms for incorporating science into decision-making are underdeveloped, and scientific practice often fails to match the scope or complexity of societal challenges. Although overall trust in science is high—and in some contexts may even be higher in the context of the COVID-19 pandemic [47]—distrust often exists among groups or in contexts where science is perceived as producing evidence that diverges from lived realities or prescribing action that fails to address local priorities [48].

Addressing these issues requires shifting our understanding of the nature of reality and how we can know it. An alternate, more useful understanding of complex systems can be achieved by methodologies which are not less rigorous but more qualitative, allow ambiguity, and perceive complexity as emerging from the interactions among system components and with the broader environment—including interactions with the observer, who is thus internalized to the system. Such an approach suggests that the complex systems we observe and are part of are amenable to understanding and thus to forecasting and influence. A critical difference from the classical approach is the observation that science itself is not indifferent to changing environments and changing decision making situations [49].

Apart from model-dependent realism, another response has been to recognize the importance of a post-normal and mission-oriented approach to science [50,51]. While normal science claims its credibility from avoiding bias, post-normal science makes a conscious choice about the biases and values most appropriate to the task of guiding science to create knowledge for practice. It likewise embraces a broader section of society (an extended peer community) in the process of producing knowledge [52]. Among problems addressed by post-normal science are those relating to major technological hazards or environmental pollution. For such issues, ethical judgements and values play as important a role as formal

analysis. Wider adoption of this epistemological stance—through conscious promotion by scientific authorities, in education, and in policy and practice—would improve our ability to meet increasingly complex challenges.

## 2. Sharpening the tools of the trade: shifting scientific methodology

To overcome persistent barriers to human understanding and effective practice, the scientific method, as typically implemented, must expand in its conceptualization and application to incorporate new methodologies; for example, systems approaches and transdisciplinary research are widely recognized as key methods for addressing complex challenges [9,29,53]. In the urban health context, Newell and Siri argue for the application of low-order system dynamics models in urban health policymaking [54].

Indeed, the practice of science is itself a complex system which would benefit from the insights of complexity science, continually adapting and honing its methods, rules, and perceptions to meet the enduring demands of a complex reality.

Similarly, to meet the demands of our new model-dependent realism, science must incorporate new domains of evidence, including practical and experiential knowledge and insights from the social sciences, and new partners from outside the science domain.

We have begun to see changes in the structure of science that would support methodological innovation. Indeed, "In every age, science is shaped around its leading problems, and it evolves with them" [55]. And thus, in the contextual shift from a 'small world on a big planet' to a 'big world on a small planet' [56], with vastly more (and more interconnected) people, cities, and other human artifacts, we have begun to see a softening of disciplinary boundaries and the appearance of hybrid scientific domains "in which consilience is implicit" [46], as well as the emergence of action-oriented, post-normal science in contexts where uncertainties are epistemological or ethical and decision stakes reflect conflicting purposes among stakeholders.

The continued expansion of novel, complexity-oriented methodologies will require explicit institutional and financial support as part of a project of field-building. It will require a serious effort by researchers to limit jargon and provide useful explanations and straightforward solutions for complex problems—without shortchanging irreducible complexity. Perhaps most of all, it will require an unflagging focus on the creation of actionable knowledge relevant to end users, given that the translation of knowledge into action is entirely dependent on societal trust in science.

Various resources are available to support this transition. To name just two, the International Science Council's global science programme on Urban Health and Wellbeing has over the last decade elaborated a systems approach to urban health and suggested actions for future research and action [57]. Meanwhile, the OECD has promulgated recommendations for supporting transdisciplinary research to address complex societal challenges, part of a wider, growing recognition of the value of this approach [9].

## 3. Translating knowledge into justified action: shifting rationality

When confronted with uncertainty (as opposed to calculable risks), limited time, data, and computational capacity—as is often the case in complex decision-making situations—traditional notions of rational action, premised on optimal decisions made under perfect information, are inaccurate and will typically fail to deliver desired outcomes. Rather, in such situations, ecological rationality, which accounts for context, and collaborative rationality, which seeks to progressively approach better answers through a process of co-discovery, should be favored.

Among other features, such rubrics may make use of heuristics to avoid the need for the explicit computation of probabilities and should be better adapted to the structure of their environment. In the shift from deterministic to probabilistic to heuristic decision-making, unrealistic assumptions of unbounded rationality and unlimited computational capabilities are increasingly relaxed. While heuristics are not a substitute for in-depth investigation, in an uncertain world, a simple heuristic—i.e., a fast, frugal, generally experience-based decision-making strategy that focuses on a small set of highly relevant

predictors—can be more accurate than other decision-making strategies when action is urgently needed. Ecologically rational decisions can be achieved when heuristics are well-matched to their environments.

Heuristics have been promoted as one approach to managing complexity, for example, in the implementation of green infrastructure [58] and in data-driven urban design [59]. Critical systems thinking in ICT4S (Information and Communication Technology for Sustainability) requires a critical systems heuristics (CSH) approach, as demonstrated in Sidewalk Labs 'smart city' project in Toronto. CSH recognizes the limits of computational approaches in urban decision making and balances the tension between facts and values which must be taken into account and the notions of how people think human life should be organized [60]. Similar examples include the partial systems analysis of noise pollution reduction in a district of Beirut [61].

Another key element of rational decision-making in complex situations is collaboration among diverse partners. As is the case for methodological approaches to knowledge production, diversity in experiences, values, priorities, and background knowledge is critical to ensuring that decisions are both well-matched to needs and are likely to achieve the buy-in necessary for effective implementation.

All complex urban decision-making situations involve a range of interacting subjects and objects and diverse types of resources, including private, common pool and public goods and services. Adequate responses will incorporate a diverse set of methods based on different rationalities, accounting for both facts and values, as well as experiences and past knowledge of the past and future needs and aspirations.

## 4. Supporting a complexity-informed approach: shifting governance

When decision making strategies become more ecologically rational, and therefore, by definition, better matched to their social and biophysical environments, they will necessarily become more inclusive, deliberative, and reflexive. The norms and principles on which decisions are based should be adaptive and not persist irrespective of the circumstances under which they were formulated, which may have changed. In such contexts, governance for complex decision-making situations will be more robust and yet more flexible than structures based on individual rationality (i.e., based on an assumption of fixed principles and relationships).

Robust governance structures contribute to health-promoting urban complexity. They are adaptive insofar as they benefit from existing decision-making rules and strategies while retaining the capacity to change by learning. They are reflexive in the sense that they are able to change themselves in response to reflections on their own performance within the changing environments they operate in [62–64].

These shifts do not happen automatically and therefore we need substantially more research on the ways in which governance mechanisms and institutional structures interact with complex systems, as well as advocacy to promote those systems that generate healthy complexity (without creating unproductive chaos).

## 4. Conclusions

Urbanization continues to create a range of challenges and inequities which threaten health and wellbeing. A new urban health science is needed to understand why and how this is happening and to adequately and rapidly respond to systemic risks and prevent human disasters. The increasing complexity that comes with urbanization does not inexorably lead to an urban advantage.

The complexity of urban systems needs to be actively shaped; urban health challenges must be met with a diversifying array of human- and social capital supported by well-functioning and well-connected physical, institutional, and technological infrastructure.

In the practice of science, coping with complexity requires explicit efforts and investments into multidisciplinary, interdisciplinary, transdisciplinary, and mission-oriented science. Due to persistent incentive structures in science, specialized, narrow knowledge published in high-ranking journals is favored over more general and practical guides,

reports or knowledge communication. These features of science practice lead to a fragmentation of knowledge and an increasing inequality of voices operating at the level of policymaking.

In terms of policy-making, coping with complexity requires more investments in the empowerment of the public, building social cohesion and participatory decision-making, not merely securing privileges and decision-making power. It requires leadership which does not shy away from complexity and is willing to address institutional and policy reforms which are tailored towards building collective action and the public good, also, or particularly, when that means that the political system needs to reform itself in order to better cope with complex decision-making situations.

In society, it means to actively engage in and with the urban health challenges that urban residents are confronted with, and for them to be part of the urban planning and design processes. It requires the articulation and recognition of people's visions of the city they need and want.

Bringing all of these required shifts together and building new interactions and alliances can be done by learning from cities in the real world which have grown and transformed successfully in response to the challenges they faced in the past. By exploring new possibilities and spaces provided by digitalization, today we are also able to take part in modelling, building and simulating healthy cities of and for the future. Complexity leadership theory [65], collaborative modelling and urban planning, citizen science, and creative collective intelligence [66] are examples which are part of a bundle of actions taking us forward to creating a new urban health science.

**Author Contributions:** All authors contributed equally to the manuscript. All authors have read and agreed to the published version of the manuscript.

**Funding:** This opinion piece is the outcome of the International Science Council's (ISC) global science programme on Urban Health and Wellbeing: A Systems Approach, which received funding by the Chinese Academy of Sciences, the Chinese Association of Science and Technology and the Xiamen City Bureau of Science and Technology.

**Conflicts of Interest:** The authors declare no conflict of interest.

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
