# Peer review of "Towards a New Urban Health Science"

_urbansci, doi:10.3390/urbansci7010030_

Round 1
Reviewer 1 Report
The authors make a case for a new methodological and systemic approach to the discipline and study of urban health science. It is a review article which offers the authors' perspective on a revised science-based approach to urban health and argues for the rationality and benefits of this approach. Contextually, the authors should make an effort to present quantitively the burden of the urban health crisis as part of the explanation/justification for new methods for shifting the scientific methodology.
Methods: the methods section of this paper does not adequately describe the review process undertaken and an analysis of the literature.
Results: Inclusion of an illustration/diagram of the proposed new approach may be very instructive for readers (i.e. systemic connections, urban complexity networks)
A proposition of a model of consilience for urban health (recommended with diagram) would be strongly indicative of the systemic inter- and trans-disciplinary inquiry posited. This would demonstrate the practical application of a 'graduation' in the scientific enterprise for urban health and the benefits thereof.
Author Response
We provided a detailed response to each of the reviewers points in the file attached.

Reviewer 2 Report
The commentary introducing a new perspective on urban health is a well-rationalized, well-articulated, and well-written paper. The paper presents various challenges in decision-making in urban health sciences that utilize a top-down approach and describes a paradigm shift toward adopting more complex thinking characterized by bottom-up and/or participatory processes in research, policy, and practice. The issue addressed in the paper is timely and relevant and contributes significantly to a new body of literature in urban sciences, especially in the context of crisis like covid -19 pandemic.
The authors bring forth a new understanding of urban health by incorporating the aspect of urban complexity, through participatory processes, in designing programs and policies to improve human health and the environment they live in.
A few observations which can be considered for the manuscript is described below.
· The Social Capital concept mentioned in the conclusion was never discussed in the main section of the manuscript. The broad framework for community engagement, qualitative and quantitative inquiry, and choosing the best practices from other countries would have helped to improve the urban health infrastructure-related policies, especially in low-income countries and low and middle-income countries.
· The bottom-up approach would consider the user perspectives however, it is unclear how the countries with non-democratic governance adopt this approach. It is also unclear how we differentiate between a health emergency and otherwise to decide on adopting the top-down or bottom-up approach.
· Unless the qualitative enquiry is objective, there is a chance of bias due to personal interests and ulterior motives of the policymakers and leaders and their ideological positions. Understanding how diverse low-income, low and middle-income, and high-income status affects their policies would help policymakers to make decisions that are consistent with the existing evidence and cultural context.
· Convincing the policymakers about the complexity and the need for it is a major challenge, especially when the political priorities are different, at least in low-income countries. The user-driven methodology is ideal, and practice may be difficult as the cities lack broad vision and space and resources. The mere copying of urban policies of other countries often incurs unnecessary burdens on the people without solving cities’ health infrastructure issues. Therefore, decision-making should be democratic and grounded in local realities.
· However, understanding the varied issues and problems from a more qualitative perspective would guide the research, policies, and governance, provided the concerned people are genuinely interested in the welfare of the entire society.
Author Response
We provided a detailed response to the reviewer in the file attached.

Reviewer 3 Report
The paper focuses on a very important and contemporary issue. Nevertheless, its analysis, although following a logical and evidence-based approach, needs more depth.
To improve the manuscript, it needs more analysis of what is presented and should also be combined with more examples.
In other words, beyond the theoretical opinions there should be examples, either of policies or of specific areas that follow these policies.
Author Response
We provided a detailed response to reviewer 3 in the file attached

Round 2
Reviewer 1 Report
Authors comments/response seeks to justify their approach to this paper and should be included in the commentary
Author Response
Thank you for the feedback regarding the justification of our approach. As advised we will include the key aspects into the commentary to clarify.
The english language will be checked by a native language speaker before final submission.
Reviewer 3 Report
The article is much better now.
Author Response
Thank you for the reviewer's feedback. We will have the article checked by a native language speaker before next submission.